# The Role of Outer Membrane Proteins in UPEC Antimicrobial Resistance: A Systematic Review

**DOI:** 10.3390/membranes12100981

**Published:** 2022-10-10

**Authors:** Inês C. Rodrigues, Sílvia C. Rodrigues, Filipe V. Duarte, Paula M. da Costa, Paulo M. da Costa

**Affiliations:** 1Laboratório de Microbiologia e Tecnologia Alimentar, Departamento de Produção Aquática, Instituto de Ciências Biomédicas Abel Salazar (ICBAS), Rua de Jorge Viterbo Ferreira, 228, 4050-313 Porto, Portugal; 2Pharmaissues, Consultoria, Lda, Rua da Esperança n° 101, Ribeira de Frades, 3045-420 Coimbra, Portugal; 3Centro de Neurociências e Biologia Celular (CNC), Faculdade de Medicina, Pólo 1, Universidade de Coimbra, Rua Larga, 3004-504 Coimbra, Portugal; 4Microbiology Department, Centro Hospitalar Universitário do Porto, Largo do Prof. Abel Salazar, 4099-001 Porto, Portugal; 5Interdisciplinary Centre of Marine and Environmental Research (CIIMAR), Terminal de Cruzeiros do Porto, de Lexões, Av. General Norton de Matos s/n, 4450-208 Matosinhos, Portugal

**Keywords:** antimicrobial resistance, bacterial proteins, outer membrane proteins, uropathogenic *Escherichia coli*

## Abstract

Uropathogenic *Escherichia coli* (UPEC) are one of the most common agents of urinary tract infection. In the last decade, several UPEC strains have acquired antibiotic resistance mechanisms and some have become resistant to all classes of antibiotics. UPEC outer membrane proteins (OMPs) seem to have a decisive role not only in the processes of invasion and colonization of the bladder mucosa, but also in mechanisms of drug resistance, by which bacteria avoid killing by antimicrobial molecules. This systematic review was performed according to the PRISMA guidelines, aiming to characterize UPEC OMPs and identify their potential role in antimicrobial resistance. The search was limited to studies in English published during the last decade. Twenty-nine studies were included for revision and, among the 76 proteins identified, seven were associated with antibiotic resistance. Indeed, OmpC was associated with β-lactams resistance and OmpF with β-lactams and fluoroquinolone resistance. In turn, TolC, OmpX, YddB, TosA and murein lipoprotein (Lpp) were associated with fluoroquinolones, enrofloxacin, novobiocin, β-lactams and globomycin resistances, respectively. The clinical implications of UPEC resistance to antimicrobial agents in both veterinary and human medicine must propel the implementation of new strategies of administration of antimicrobial agents, while also promoting the development of improved antimicrobials, protective vaccines and specific inhibitors of virulence and resistance factors.

## 1. Introduction

Uropathogenic *Escherichia coli* (UPEC) are the most common agents of urinary tract infection (UTI), in both humans and pets [1,2]. It is estimated that UPEC are responsible for more than 80% of UTIs in humans and between 30 to 69% of UTIs in pets [1,3]. A successful combination of virulence factors confers to these strains an increased capacity of ascending through the urinary tract, colonizing, invading and disseminating in the bladder mucosa. From there on, they can continue to progress to the kidneys (pyelonephritis) as well as enter the bloodstream, causing bacteremia [4]. UPEC strains are mainly classified into virulent-phylogroup B2, possessing specific and diverse virulence factors responsible for colonization, invasion and dissemination; to a lesser extent, they are also included in phylogroup D [1,5].

The treatment of UTI relies on antibiotic therapy, such as trimethoprim-sulfamethoxazole, fluoroquinolones and cephalosporins, representing the first-line empirical antibiotics [1,6,7]. According to the European Center for Disease Prevention and Control (ECDC), more than half (58.3%) of the *E. coli* isolates reported in 2018 were resistant to at least one of the antimicrobial groups (i.e., aminopenicillins, fluoroquinolones, third-generation cephalosporins, aminoglycosides and carbapenems) [8]. The dramatically increasing rate of multidrug-resistant strains carries a higher risk of treatment failure, entailing increased costs in health care [2,9]. In fact, resistance to carbapenems, a class of antibiotics used to treat some of the most severe bacterial infections, and resistance to colistin, also a last-line antibiotic, have been described in UPEC isolates [10,11].

Despite a deepening awareness of antimicrobial resistance selection among companion animals, many antimicrobials paramount for human health are used in pets [12]. As people and companion animals share routines and living spaces, the circulation of multi-resistant UPEC strains between companion animals and their owners is of concern, as these strains carry with them virulence and resistance factors [12,13]. Indeed, multiple human associated extended-spectrum cephalosporin-resistant UPEC strains have been isolated from cats and dogs, suggesting clonal dissemination [1].

Whenever antimicrobials are used, bacteria inevitably develop resistance mechanisms, including the modification of proteins of the outer membrane (for example, the loss of porins) [14]. Among Gram-negative bacteria, as is the case of *E. coli*, the presence of the outer membrane is key feature, conferring a crucial and impermeable barrier to the passage of toxic chemicals, such as antibiotics [15,16]. The outer membrane is composed by phospholipids, lipopolysaccharides (LPS), and a myriad of proteins (outer membrane proteins—OMPs), from which OMPs represent approximately half of the cell wall of Gram-negative bacteria [17]. OMPs are responsible for several functions, such as antibiotic and iron transportation, host mucosal adhesion and membrane integrity [5].

OMPs of UPEC, such as flagella, fimbriae, porins, iron receptors and efflux pumps, possess distinctive features, which allow them to invade and colonize the bladder mucosa, representing a crucial tool for both UTI development and antimicrobial resistance. Therefore, since the OMPs are on the bacterial surface, they represent critical targets for the development of improved antimicrobials, protective vaccines and new therapeutic strategies [18].

Through an extensive systematic method, this review aims at an in-depth characterization of the proteins of the outer membrane of UPEC, highlighting the OMPs and their role in antimicrobial resistance.

## 2. Materials and Methods

This systematic review was performed according to the relevant points of the PRISMA (Preferred Reporting Items for Systematic Reviews and Meta-analyses) guidelines [19].

### 2.1. Selection Strategy

In September 2022, an independent researcher (I.C.R.) searched the PubMed database without language restrictions in the past ten years (since 2012). Peer-reviewed studies describing antimicrobial susceptibility patterns in terms of proteins located in the outer membrane of UPEC isolated from human or pet samples of any age and region were included in this review. Studies published in a language other than English or Portuguese, publications comprising editorials, comments, letters to the editor, guidelines, theses, books and scientific meeting abstracts, literature reviews or case reports, studies performed specifically in ESBL and/or related to other bacteria and that did not mention uropathogenic *E. coli* or UTI in the title, publications without a description of outer membrane proteins, and studies published before 2012 were excluded. The author (I.C.R.) also reviewed the reference lists from the review articles reported in the PubMed and Web of Science searches to identify possible additional articles for inclusion. A combination of the following search terms was used: Uropathogenic *Escherichia coli* AND membrane proteins. In order to evaluate a potential inclusion in this study, titles and search results were examined.

### 2.2. Selection Process and Data Extraction

All search results were exported to Microsoft Office™ Excel. Results from the initial search were evaluated separately by the two review authors (I.C.R and S.C.R.) according to the inclusion criteria. First, the results were screened by reading the article titles and excluding articles that were not relevant according to the inclusion criteria. Afterwards, the study abstracts were evaluated and non-relevant articles were excluded. Finally, the full-text articles selected by the two reviewers were collected and assessed for their relevance relative to the inclusion criteria. Any disagreements regarding the eligibility of studies were reconciled at the final step by discussion and consensus.

Upon the final consensus, the following data were extracted from each selected study and validated by the second author (S.C.R.) elaborating a systematic database:Title, authors and outer(s) membrane(s) detected;Samples origin and type;Method used for the detection of outer membrane protein;Group of outer membrane protein;Description of outer membrane protein;Function of outer membrane protein.

### 2.3. Quality Assessment

Risk of bias (RoB) was examined using the Newcastle–Ottawa scale (NOS) quality assessment scale for cohort studies [20]. The used scale was adapted from the NOS and the assessment was based on three criteria: sample (maximum of 3 points), comparability (maximum of 2 points) and outcome (maximum of 3 points). This was done by I.C.R. and S.C.R. and three categories were decided: low RoB (rating 7 to 8 points), moderate RoB (rating 5 to 6 points) and high RoB (rating from 0 to 4 points). No article was excluded based on this assessment.

## 3. Results

### 3.1. Description of Studies

The PRISMA flow diagram summarizes the number of records screened and included (Figure 1). The characteristics and methodological quality of the included studies are presented in Table 1. The literature search using PRISMA identified a total of 1122 studies. After removing the duplicates, 569 were screened for eligibility. After the screening of titles and abstracts, 520 studies were excluded. Full texts of the remaining 48 studies were read and 22 more studies were excluded. At the end, 29 publications were included in this systematic review.

### 3.2. Quality Assessment

Based on the quality assessment of studies using the NOS assessment, three studies scored 8 points [14,17,25], which could be regarded as good studies. Another fourteen recorded 7 points, being classified as low Rob [4,9,18,22,23,24,26,31,33,34,37,38,40,42]. Lastly, twelve studies scored 5–6 points [5,6,10,21,27,28,29,30,32,35,39,41], and could be regarded as satisfactory studies.

### 3.3. Characteristics of Studies and Outcomes Measures

Characteristics of the included studies are presented in Table 1. In a total of 29 studies, 24 encompassed only human isolates [4,5,6,9,10,17,18,21,22,23,24,25,26,28,29,31,33,34,35,36,37,38,39,40,41], one study comprised human and avian isolates [32] and one regarded only dog isolates [14]. Among the first group, seven studies included urine isolates [4,17,35,36,38,39,40]; fourteen comprised urine and blood isolates [5,6,9,10,18,23,24,26,28,29,31,33,34,37]; three studies involved urine, blood and fecal isolates [21,22,25]; and one study addressed fecal, urine and vaginal samples as well as cerebrospinal liquid. Regarding animal samples, only one study evaluated dog samples [14] and a different one included avian fecal samples along with human isolates [32].

Considering the methodology used to identify directly the OMPs of UPEC strains, sodium dodecyl sulfate–polyacrylamide gel electrophoresis (SDS–PAGE) was performed in eleven studies [4,17,18,22,25,26,31,33,34,38,40], Western blot (WB) in eight [4,22,25,26,34,38,40,41], nano-liquid chromatography tandem mass spectrometry (nLC-MS/MS) in three [5,14,37], liquid chromatography coupled to tandem mass spectrometry (LC-MS/MS) in two [5,24], two-dimensional difference gel electrophoresis (2D DIGE) in two [14,18], fast protein liquid chromatography (FPLC) in one [30] and tandem mass spectrometry in one [18]. In order to evaluate the function of a specific OMP and/or their indirect presence, adhesion methods were performed in seven [21,28,29,31,37,38,40], biofilm assays were performed in six studies [31,33,37,39,40,41], hemagglutination assays (HA) in three [28,29,31], an invasion method in four [4,28,29,40], crystallization and structure determination in three [10,24,27], immunofluorescence imaging in three [10,40,41], proteolytic activity in three [22,25,29], autoaggregation assays in two [40,41], immunoblot analysis in two [21,33], membrane integrity assays in two [22,23], motility assays in two [31,33], bacteriophage adsorption assay in one [6], a chelator assay in one [6], dose response analysis in one [6], electron microscopy in one [28], an extracellular matrix binding method in one [41], high-throughput screen (HTS) in one [6], in silico docking in one [10], molecular docking studies in one [35], a nematode killing assay in one [33], a polymorphism study in one [9], survival studies in one [4] and transmission electron microscopy (TEM) in one [37].

### 3.4. Characterization of Outer Membrane Proteins of UPEC

Several proteins were identified as constitutively part of the UPEC strains outer membrane. One study described the role of the BamA protein [21], nine studies described fimbriae and their function [4,5,10,24,27,28,37,39,40] and another ten characterized the porins found in UPEC isolates [5,14,17,22,23,25,29,32,33,35]. Seven studies described the role of lipoproteins [4,5,6,18,26,31,34], three studies described the importance of efflux pumps [33,35,36], three studies identified iron receptors and siderophores in the outer membrane of UPEC [5,6,18], one study identified TosA protein and its role [38] and one study identified Traf and Ydef [5]. In addition, two studies described the importance of flagella in UPEC motility, encoded by flagellin (FliC) [4,31], and three studies characterized the role of the phase-variable antigen 43 autotransporter protein (Ag43) [5,40,41].

ChuA, FepA, FyuA, NmpC, OmpA, OmpC, OmpF, OmpT, OmpX and SlyB proteins were identified in more than 80% of UPEC strains [5]. According to Wurpel et al. [5], Omp A, OmpX and OmpC were present in all UPEC isolates, while OmpT, NmpC and OmpF were identified in 96%, 89% and 83% of UPEC isolates, respectively. Dehghani et al. [17] have also identified OmpA and OmpC as the most prevalent proteins in the outer membrane of the studied UPEC.

All of the identified proteins have an essential function as described in Table 2. The functional diversity of the OMPs ranges from motility and adhesion (flagella, fimbriae, adhesins) to survival within the urinary tract (iron receptors, siderophores and efflux pumps) [4,5]. Their individual or joint action in the host tissue entails a high level of virulence and pathogenicity, leading to severe UTI symptoms [33,35,36,39].

### 3.5. Association to Antimicrobial Resistance

Association to antimicrobial resistance was found for TosA, TolC, Lpp, OmpC, OmpF, OmpW and YddB [14,17,23,32,33,34,35,36,38]. Xicohtencatl-Cortes et al. [38] found that TosA, a nonfimbrial adhesin, binds to host epithelial cells receptors from the upper urinary tract, contributing to the pathogenesis of UPEC. The same authors suggested that UPEC strains producing TosA exhibited a strong association and antimicrobial resistance to β-lactam and non β-lactam antibiotics, such as penicillin, β-lactamase inhibitors and inhibitors of the folate pathway [38].

Overexpression of the AcrA–AcrB–TolC efflux pump complex (QepA and OqxAB) increases antimicrobial resistance of UPEC strains towards β-lactams (mainly cefoxitin), chloramphenicol, cyclines and, notably, quinolones [33,35]. Pantel et al. [33] suggested that the increased efflux capacity of *E. coli* strains involves several efflux pumps from different families, as well as other genes encoding efflux systems less frequently described in clinical resistance (acrEF, mdfA, yhiV, acrD and tehA) that were highly overexpressed.

According to Tavio et al. [36], the AcrA–AcrB–TolC efflux pump is the major multidrug efflux transporter in *E. coli*, allowing the passage of fluoroquinolones, among other antimicrobials.

Pantua et al. [34] found that the deficient production of Lpp leads to an increased OM permeability, a leakage of periplasmic components and an increased outer membrane vesicle release. The same authors also described that complete deletion of Lpp causes globomycin resistance [34].

Among the identified porins, OmpC, OmpF, OmpW and YddB were shown to be the preponderant in antimicrobial resistance strategies of UPEC strains. OmpC is involved in both the transport of small molecular weight hydrophilic materials across the outer membrane and in antimicrobial resistance (e.g., β-lactams antibiotics) [17,32]. OmpF plays an important role in antibiotic transport, such as β-lactams and fluoroquinolones [23]. In fact, a decreased *ompF* expression was related to an increase of resistance towards β-lactam antibiotics in *E. coli* [23,35], since OmpF is the main gate for enrofloxacin entrance [14]. Lastly, YddB (a putative porin protein) seemed to be the major porin involved in the passive transport of novobiocin across the outer membrane [23].

## 4. Discussion

UPEC is the most frequent uropathogen worldwide, entailing elevated healthcare costs and a high risk of treatment failure in both human and veterinary medicine due to the emergence of multidrug-resistant strains and limited therapeutic options to treat infections caused by these strains.

Just as a One Health unifying approach has the merit of reframing our understanding about the capacity of resistant bacteria to spread between different biomes [42,43], a wide overview of resistance mechanisms used by bacteria to evade antimicrobial treatment will be crucial for overcoming the problem and to find both new drug targets and new structural classes of antibiotics. Herein, among 76 OMPs described in the 29 studies included in this systematic review, seven were responsible for resistance to several antimicrobial drugs. Among them, OmpC is associated with β-lactams resistance and OmpF with β-lactams and fluoroquinolones resistance. With a prevalence of less than 75%, TolC, YddB, TosA and Lpp are also associated with resistance against fluoroquinolones, enrofloxacin, novobiocin, β-lactams and globomycin, respectively. Considering that fluoroquinolones and β-lactams are the antimicrobial classes more commonly used for the treatment of UTI in both human and animals, the spread of these OMPs between UPEC strains is of enormous clinical relevance. Indeed, both antimicrobial drugs are particularly affected by changes in the permeability of the outer membrane of UPEC strains, since β-lactams and fluoroquinolones often use water-filled diffusion channels (porins) to penetrate this external barrier [44]. Despite the predominant mechanism of resistance to β-lactams in UPEC being mainly attributed to the production of β-lactamases [45], OMPs seem to possess essential complementary “tools”, as they control the entry of these molecules to the periplasmic space, allowing the production of β-lactamases in sufficient concentrations to achieve the destruction of the antibiotic molecules [46].

Porins are outer membrane β-barrel proteins that cross the membrane cell, responsible for uptake of nutrients, being essential for bacterial survival and growth [33]. On the other hand, they allow the “unwanted” entrance of several antimicrobial molecules; for example, OmpC, OmpF and OmpW are porins responsible for creating a size-selective channel, promoting the transport of small hydrophilic molecules, including β-lactams and fluoroquinolones [47]. Thus, the production of OmpW and OmpC was reduced in tetracycline and nalidixic acid resistant strains, indicating an antibiotic-specific pattern of porin expression [47,48]; and a decreased *ompF* expression was related to an increase of resistance towards β-lactam and fluoroquinolones antibiotics [23,35], since OmpF is the main gate for enrofloxacin entrance [14]. In addition, OmpF and OmpC production seemed to be influenced by the environment of UPEC during a UTI. Urine osmolarity appears to affect *ompF* and *ompC* expression: *ompF* is upregulated under a low osmolarity condition, while *ompC* is upregulated under high osmolarity [49]. Although it is a lesser reported porin, YddB is associated with passive transport of novobiocin across the outer membrane. Presently, novobiocin resistance is not an emerging problem; however, the mechanism by which YddB confers novobiocin resistance has not yet been characterized.

Beyond the alteration of membrane permeability, bacteria is also able to enhance its resistance by the overproduction of efflux pumps (actively expelling the antibiotic out of the cells) [50]. Efflux pumps may be the most rapid acting and effective resistance mechanism in the bacterial defense [50]. Regarding the AcrA–AcrB–TolC efflux pump, Chowdhury et al. [51] hypothesized that one of the intrinsic mechanisms of multidrug-resistant UPEC strains is the overexpression of the AcrAB–TolC efflux pump, probably caused by the mutation of the acrR gene.

Among the OMPs related to antimicrobial resistance, the TosA protein, a nonfimbrial adhesin, is a virulence marker in UPEC strains, since deletion of the tosA gene affected their capacity to colonize the bladder and kidney in a murine model [52,53]. Moreover, multidrug-resistant UPEC isolates expressing the tosA gene maintained a high correlation with virulence genes presence [53]. However, additional research is required to fully understand the role of TosA during UTI.

Another OMP associated to antimicrobial resistance is Lpp, which is the most abundant lipoprotein of *E. coli* (more than 5 × 10^5^ molecules per bacterial cell), being responsible for maintaining the stability and integrity of the outer membrane [54]. Although Pantua et al. [34] described that the complete deletion of Lpp led to globomycin resistance, other authors stated that the deletion of this lipoprotein increased the susceptibility for vancomycin, erythromycin and rifampicin antibiotics [54,55]. In fact, the reduction of Lpp stimulated vesicle formation, decreased membrane integrity and reduced cell invasiveness by turning down biofilm formation [54,55,56].

In addition to the role of the OMPs, other proteins are also involved in the antimicrobial defence of bacteria. Qnr proteins protect DNA gyrase and type IV topoisomerase IV, being responsible for quinolones resistance in UPEC isolates [35]. These proteins supplement resistance to quinolones by efflux pump activation, altered quinolone target enzymes or deficiencies in OMPs [57]. RecA protein also participates in DNA reparation and activation of the bacterial SOS system [35]. At last, Spr (peptidoglycan endopeptidase) proteins assists peptidoglycan biogenesis in UPEC and flagella expression, which are correlated to a fitness advantage during bladder colonization [4].

Given the potential zoonotic of UPEC, the origin of the isolates of each study was also assessed. Although most of the isolates included in this review were from human origin, UPEC strains are being able to circulate among human and pets populations [13]. Belanger et al. [12] corroborated the phylogenetic proximity between human isolates and those obtained from cohabitant cats and dogs. Thus, we questioned whether the OMPs of UPEC isolated from pets would be different from those isolated from human origin. Piras et al. [14] suggested that if UPEC isolated from humans are exposed to the same concentration of antibiotic that animals are exposed to, then the mechanisms involved in resistance will be similar. Moreover, it requires further characterization of OMPs in UPEC isolated from both human and pet samples, as well as the elucidation of the mechanisms involved in antimicrobial resistance.

UPEC strains have cellular membrane machinery that allows them to be very effective when infecting the urinary system. Combining the fact that *E. coli* possess both a great ability of adaptation to different environments and present highly conserved proteins and organelles, a rapid acquisition of the tools necessary for antimicrobial resistance is promoted. Since available antimicrobial options have become ineffective for treatment, the access to antimicrobial molecules should be rethought and empirical therapy should be avoided. Therefore, the emergence of antimicrobial resistances is an important public health problem. Without implementing more stringent measures, the emergence of “more” multi-resistant strains will occur, leading us to a point of no return.

New therapeutic alternatives need to be discovered to combat the emergence of multidrug UPEC as well as other pathogens [58,59]. Several antimicrobial molecules that interact with OMPs have been already studied, such as darobactin, Polyphor peptide 7 and MRL-494, which affect the integrity of the bacterial outer membrane by inhibition of BamA function [60]. Pilicides and curlicides compounds have the ability to inhibit the pili of UPEC [61,62]. Pre-existing molecules used for other applications may also have antibacterial potential against UPEC, such as nitazoxanide, which is used for intestinal parasitic diseases and also inhibits the function of type 1 and Pap pili [63]. Therefore, the OMPs described in this review seem to be crucial targets in the fight against UPEC and they may also be key targets for sparing other multidrug-resistant pathogens.

This systematic review has several limitations that should be considered. Firstly, the enormous variety of methodology used and often not directly detecting the protein, but rather its effect, may create some bias. Secondly, six published studies were not included as full text articles were not accessible online; thus, some important data on OMPs and antimicrobial resistance may be missing. Thirdly, the isolates included in this review were mostly from human origin; therefore, a comparison of prevalence, function and role in antimicrobial resistance of OMPs isolated from humans and pets was not possible, skewing the research towards humans irremediably. Despite limitations on reporting, the strengths of this review are the low prevalence of high RoB studies and the broader understanding of the potential antibiotic resistance for different outer membrane proteins.

This systematic review highlights the need for further investigation about the role of OMPs in antimicrobial resistance among UPEC, as well as the prudent use of antimicrobial agents in both veterinary and human medicine.

## 5. Conclusions

In conclusion, our findings showed that several OMPs are related to antimicrobial resistance. ChuA, FepA, FyuA, NmpC, OmpA, OmpC, OmpF, OmpT, OmpX and SlyB were identified in more than 80% of UPEC. However, only OmpC, OmpF, TolC, OmpX, YddB, TosA and Lpp are related with antimicrobial resistance. The classes of antibiotics most affected by antimicrobial resistance conferred through OMPs are fluoroquinolones and β-lactams. These results reflect the urgency of the implementation of new strategies of administration of antimicrobial agents in both veterinary and human medicines, in order to subvert the emergence of multidrug-resistant UPEC strains.

## Figures and Tables

**Figure 1 membranes-12-00981-f001:**
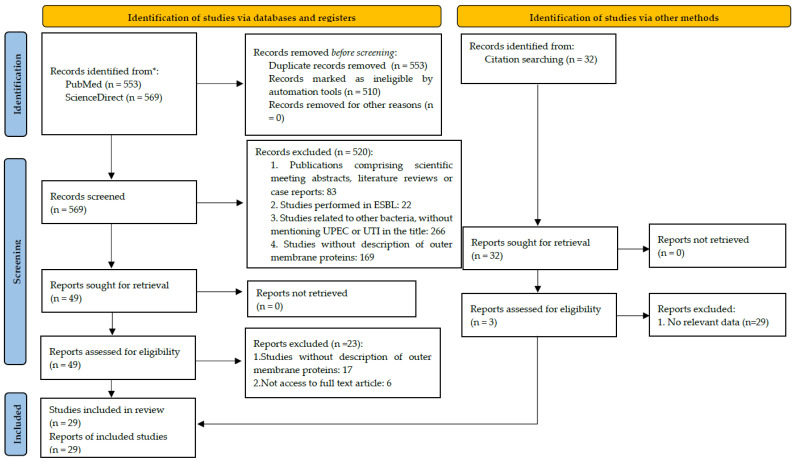
PRISMA 2020 flow diagram. * Consider, if feasible to do so, reporting the number of records indenified from each database or register searched (rather than the total number across all databases/registers). Adapted from [19].

**Table 1 membranes-12-00981-t001:** Characteristics of included studies after full assessment. Data from the final twenty-nine studies were extracted and systematized. Author; year of publication; the species of the isolates and the type of the sample collected; and the method used for protein characterization and the OMP detected were listed. The RoB assessed through the adaptation of NOS quality scale was also included for each study.

Author	Year of Publication	Origin of Isolates	Sample Type	Method	Risk of Bias	OMP Detected	Reference
Alteri et al.	2008	Human	Urine and blood	2D-DIGE, MS/MS, SDS-PAGE	Low	IutA	[18]
Beck et al.	2016	Human	Urine, blood and faecal	Adhesion assays, immunoblot analysis	Moderate	BamA OmpC OmpF	[21]
Brannon et al.	2013	Human	Urine, blood and fecal	Membrane integrity assays, proteolytic activity, SDS-PAGE, WB	Low	OmpT	[22]
Choi et al.	2020	Human	Urine and blood	Membrane integrity assays	Low	OmpA OmpC OmpF	[23]
Conover et al.	2016	Human	Urine and blood	Crystallization, structure determination and further vaccination of an infected mouse	Low	FimIH	[24]
Dehghani et al.	2016	Human	Urine	SDS-PAGE	Low	OmpA OmpC OmpF	[17]
Desloges et al.	2019	Human	Urine, blood and fecal	Proteolytic activity, SDS-PAGE, WB	Low	OmpP OmpT	[25]
Diao et al.	2018	Human	Urine and blood	LC-MS/MS, SDS-PAGE, WB	Low	Bam A BamD BamE c1765 c2482 c3153 chuA FimD IreA KpsD lolB Lpp nmpC OmpA OmpC OmpF OmpX Slp SurA tolC	[26]
Geibel et al.	2013	Non-specified origin	-	Crystallization, structure determination, computational methods	Moderate	Complex FimD–FimC–FimF–FimG–FimH	[27]
Ghosh et al.	2019	Human	Urine	Polymorphism study	Low	FimH	[9]
Greene et al.	2015	Human and non-specified origin	Urine and blood	Electron microscopy, HA, immunoblot analysis	Moderate	Type 1 pili	[28]
He et al.	2015	Human and non-specified origin	Urine and blood	Adhesion and invasion assays, HA, proteolytic activity	Moderate	OmpT	[29]
Hederson & Thanassi	2013	Non-specified origin	-	FPLC	Moderate	FimC–FimH chaperone–adhesin Type 1 pilus usher FimD P pilus usher PapC PapD–PapG chaperone–adhesin	[30]
Hirakawa et al.	2019	Human	Urine and blood	Adhesion and biofilm assays, HA, motility assays, SDS-PAGE	Low	TolB	[31]
Huang et al.	2020	Human	Urine	LC-MS/MS, invasion assay, macrophage and serum survival methods, SDS-PAGE, survival in SDS and low pH conditions assays, WB	Low	FliC	[4]
Kalas et al.	2018	Human	Urine and blood	Crystallization, structure determination, in silico docking, immunofluorescence microscopy	Moderate	Type 1 pili F9 pili	[10]
Nielsen et al.	2020	Human and avian	Fecal, vaginal and CSF	In silico analysis	Moderate	OmpA	[32]
Pantel et al.	2016	Human	Urine and blood	Biofilm, nematode-killing and motility assays, immunoblot analysis, SDS-PAGE	Low	AcrA-AcrB-TolC OmpC OmpF	[33]
Pantua et al.	2022	Human and non-specified origin	Urine and blood	SDS-PAGE, WB	Low	Lpp	[34]
Piras et al.	2015	Dog	Urine	2D-DIGE, nLC-MS/MS	Low	OmpW	[14]
Ribeiro et al.	2020	Human	Urine	Molecular docking studies	Moderate	QepA OqxAB	[35]
Tavio et al.	2010	Human	Urine	SDS-PAGE	Low	AcrA–AcrB–TolC	[36]
Wurpel et al.	2015	Human	Urine and blood	nLC-MS/MS	Moderate	BamA BamB BamC BtuB ChuA CirA CjrC FadL FepA FhuA FhuE FitA Fiu Flu FyuA GdhA Iha IreA IroN IutA Lpp LptD LptE MalB NlpD NmpC OmpA OmpC OmpF OmpT OmpX Pal RcsF SmpA TraT Tsh Tsx Slp SlyB UTI89_CC1129 UTI89_C4946 VacJ Wzi YbhC YddB YeaF YgeR	[5]
Wurpel et al.	2016	Human	Urine and blood	Adhesin and biofilm assays, nLC-MS/MS, TEM	Low	BtuB ChuA CirA CjrC FepA FhuA Fiu Flu FyuA IutA Iha IreA IroN NmpC OmpA OmpC OmpF OmpT OmpW OmpX UidC UTI89C2234 SlyB	[37]
Xicohtencatl-Cortes et al.	2019	Human	Urine	Adhesion assays, SDS-PAGE, WB	Low	TosA	[38]
Yep et al.	2014	Human	Urine and blood	Bacteriophage adsorption assay, chelator assay, dose-response analysis, HTS	Moderate	RecA	[6]
Zalewska-Piatek et al.	2013	Human and non-specified origin	Urine	Biofilm assays	Moderate	Dr Fimbriae	[39]
Zalewska-Piatek et al.	2015	Human and non-specified origin	Urine	Autoaggregation assay, biofilm, adhesion, invasion assays, immunofluorescence microscopy, SDS-PAGE, WB	Moderate	DraD DraE	[40]
Zude et al.	2014	Non-specified origin	-	Autoaggregation and biofilm assays, extracellular matrix protein binding method, immunofluorescence microscopy, WB	Moderate	Ag43	[41]

2D-DIGE: Two-dimensional difference gel electrophoresis; CSF: cerebrospinal fluid; HA: hemagglutination assays; FPLC: fast protein liquid chromatography; HTS: high-throughput screen; LC-MS/MS: liquid chromatography coupled to tandem mass spectrometry; MS/MS: tandem mass spectrometry; NA: not available; nLC-MS/MS: nanoscale liquid chromatography coupled to tandem mass spectrometry; SDS-PAGE: sodium dodecyl sulfate–polyacrylamide gel electrophoresis; TEM: transmission electron microscopy; WB: Western blot.

**Table 2 membranes-12-00981-t002:** Overview of UPEC OMPs and their association to antimicrobial resistance. Integrated view of results is described, listing the OMP found in UPEC’s outer membrane as well as their function.

Group of OMP	OMP	Description	Function	Association to Antimicrobial Resistance
Autotransporter	Ag43	Phase-variable antigen 43 autotransporter protein	Autotransporter	-
BAM complex	BamA BamB BamC BamD BamE	Outer membrane protein assembly factor A Outer membrane protein assembly factor B Outer membrane protein assembly factor C Outer membrane protein assembly factor D Outer membrane protein assembly factor E	OMP assembly OMP assembly OMP assembly OMP assembly OMP assembly	- - - - -
Efflux pumps	TolC	Outer membrane protein ToC	Substance efflux	Yes
Fimbriae	DraE DraD FimA FimC FimD FimF FimG FimH FimIH PapC PapD PapG	Dr fimbriae subunit Dr fimbriae subunit tip Type 1 fimbrial protein, A chain Type 1 pilus chaperone, FimC Outer membrane usher protein FimD Outer membrane protein FimF Outer membrane protein FimG Type 1 pilus adhesin, FimH Pili FimH-like adhesin P pilus protein PapC P pilus chaperone PapD P pilus protein PapG	Dr fimbriae Adhesin Type 1 pili structure Type 1 pili structure Type 1 pili structure Type 1 pili structure Type 1 pili structure Adhesin Adhesin P pili structure P pili structure Adhesin	- - - - - - - - - - - -
Flagella	FliC	Flagellin	Flagella struture	-
Iron receptor and siderophores	c2482 ChuA CirA CjrC FepA FhuA FhuE FitA Fiu FyuA Iha IreA IroN IutA TonB–ExbB–ExbD UTI89_C1129 UTI89C2234	Putative outer membrane receptor for iron or colicin Heme/hemoglobin receptor Colicin I receptor and iron receptor Putative siderophore receptor Ferrienterobactin receptor Ferrichrome–iron receptor Ferri–rhodotorulic siderophore receptor Ferrichrome iron transport receptor Catecholate siderophore recetor Yersiniabactin receptor Siderophore receptor/adhesion Putative siderophore receptor Samochelin receptor Ferric aerobactin receotor TonB system Putative heme/hemoglobin receptor Putative iron compound receptor	Metal ion receptor and transport Metal ion receptor Metal ion receptor and transport Metal ion receptor Metal ion receptor Metal ion receptor Metal ion receptor Metal ion receptor Metal ion receptor Metal ion receptor Metal ion receptor Metal ion receptor Metal ion receptor Metal ion receptor Metal ion receptor and transport Metal ion receptor Metal ion receptor	- - - - - - - - - - - - - - - - -
Lipoproteins	lolB NlpD Lpp LptD LptE Slp SlyB VacJ Wzi YbhC YgeR	Outer-membrane lipoprotein LolB Lipoprotein Murein lipoprotein Putative uncharacterized protein LPS assembly lipoprotein Outer membrane protein Outer membrane protein Lipoprotein Outer membrane protein Putative pectinesterase Lipoprotein	LPS organization/synthesis LPS organization/synthesis LPS organization/synthesis LPS organization/synthesis LPS organization/synthesis Unknown Unknown Unknown LPS organization/synthesis LPS organization/synthesis LPS organization/synthesis	- - Yes - - - - - - - -
Other	c1765 c3153 Flu GdhA KpsD RcsF SmpA TosA TraT Tsh SurA UTI89_C4946 YeaF	Partial Putative outer membrane channel protein Putative outer membrane protein of prophage Antigen 43 AT NADP-specific glutamate dehydrogenase Putative outer membrane translocon for export of group 2 capsular polysaccharides Regulator in colanic acid synthesis Small protein A Putative repeats-in-toxin protein Conjugal transfer surface exclusion protein Temperature sensitive hemagglutinin AT Outer membrane protein chaperone Putative filamentous hemagglutinin Putative LPS scaffolding protein	Unknown Unknown Autoaggregatiom Aminoacid metabolismo Transport Unknown OMP assembly Nonfimbrial adhesin Conjugation Unknown OMP assembly Unknown Unknown	- - - - - - Yes - - - - -
Porin	MalB NmpC/c2348 OmpA OmpC OmpF OmpP OmpT OmpW OmpX Tsx UidC YddB	Maltose-inducible porin Outer membrane porin protein Outer membrane protein A Outer membrane protein C Outer membrane protein F Outer membrane protein P Outer membrane protein T, protéase VII Outer membrane protein W Outer membrane protein X Nucleoside-specific channel forming protein Outer membrane porin protein Putative porin protein	Porin Porin Porin Porin Porin Porin Proteolysis Unknown Unknown Porin Porin Porin	- - - Yes Yes - - Yes - - - Yes
Tol–Pal System	Pal	Peptidoglycan-associated lipoprotein	Membrane integrity	-
Transporters	BtuB FadL	Vitamin B-12 receptor Bifunctional long-chain fatty acid transporter	Transporter Transporter	- -

(-) not reported.

## Data Availability

Not applicable.

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
