# Peer review of "The Role of Outer Membrane Proteins in UPEC Antimicrobial Resistance: A Systematic Review"

_membranes, 2022, doi:10.3390/membranes12100981_

Round 1

Reviewer 1 Report

Good review work with impact on this topic!

Hard points:

This systematic review of the Uropathogenic Escherichia coli (UPEC), the most common agents of urinary tract infection, respects the PRISMA guidelines, aiming to characterize UPEC OMPs and identify their potential role in antimicrobial resistance. Due to the avalanche of information, the search was limited to studies published during the last decade. 

Introduction: short but consistent introduces the reader to the topic of UPEC multifaceted aspects.

M&M: is well presented and performed according to the relevant point of the PRISMA (Preferred Reporting Items for Systematic Reviews and Meta-analyses) guidelines.

Results: well written, especially points "3.3. Characteristics of Studies and Outcomes Measures" and "3.4. Characterization of outer membrane proteins of UPEC".

What to improve

Being an important biblio-study I think that "Discussion" could be improved more by adding other Info related to this topic presented recently in the “main stream”.

Decision: Minor revision

Author Response

The authors acknowledge the reviewer for its careful reading and the comments provided. Recently, the mainstream topics are vast and take us in very different directions. During the discussion, we access the One Health approach, zoonotic diseases and the urge of new antimicrobial strategies (2nd, 8th and 9th paragraphs, respectively). Thus, we have considered to strengthen alternative therapies topic (with the inclusion of a new paragraph), as we think they will be a valuable solution in the fight against multidrug-resistant UPEC strains.

Reviewer 2 Report

Good review article with addition finding for many previous work in this point, the positive point is that the author mention most of this works as references, I recommended this article to be accepted with changing this keyword only. This long keyword (Bacterial Outer Membrane Proteins), change it.

Author Response

We warmly thank the reviewer for their kind cooperation in improving this manuscript. The keyword chosen was based on Mesh dictionary terms (https://www.ncbi.nlm.nih.gov/mesh/?term=bacterial+outer+membrane+proteins) where “Bacterial Outer Membrane Protein” seemed to us the most suitable word for this work. Nevertheless, we acknowledge your comment and we have changed the keyword to “bacterial proteins” and “outer membrane proteins”.